# A Low Cost and Eco-Sustainable Device to Determine the End of the Disinfection Process in SODIS

**DOI:** 10.3390/s23020575

**Published:** 2023-01-04

**Authors:** Diego Sales-Lérida, Juan Grosso, Pedro Manuel Martínez-Jiménez, Manuel Manzano

**Affiliations:** 1Department of Automation Engineering, Electronics and Computer Architecture and Networks, University of Cádiz, 11519 Cádiz, Spain; 2Department of Software Engineering, University of Granada, 18071 Granada, Spain; 3Department of Environmental Technologies, Faculty of Marine and Environmental Sciences, University of Cádiz, 11510 Cádiz, Spain

**Keywords:** lethal UV dose, safe drinking water, solar water disinfection (SODIS), sustainability, UV sensors

## Abstract

The lack of safe drinking water is one of the main health problems in many regions of the world. In order to face it, Solar water disinfection (SODIS) proposes the use of transparent plastic containers, which are filled with contaminated water, and exposed to direct sunlight until enough UV radiation is received to inactivate the pathogens. However, a reliable method for determining the end of the disinfection process is needed. Although several approaches have been proposed in the literature for this purpose, they do not strictly accomplish two critical constraints that are essential in this type of project, namely, low cost and sustainability. In this paper, we propose an electronic device to determine when the lethal UV dose has been reached in SODIS containers, which accomplishes both constraints mentioned above: on the one hand, its manufacturing cost is around EUR 12, which is much lower than the price of other electronic solutions; on the other hand, the device is sufficiently autonomous to work for months with small low-cost disposable batteries, thereby avoiding the use of rechargeable batteries, which are considered hazardous waste at the end of their useful life. In our approach, we first analyze different low cost UV sensors in order to select the most accurate one by comparing their response with a reference pattern provided by a radiometer. Then, an electronic device is designed using this sensor, which measures the accumulated UV radiation and compares this value with the lethal UV dose to determine the end of the disinfection process. Finally, the device has been manufactured and tested in real conditions to analyze its accuracy, obtaining satisfactory results.

## 1. Introduction

Lack of access to safe drinking water remains one of the main health problems in many regions of the world. According to recent reports of UNICEF and the World Health Organization (WHO) [1], in 2020 around one in four people lacked safely managed drinking water in their homes, which implies that about two billion people worldwide consume contaminated water. As shown on the map in Figure 1, there are regions in Africa and Southeast Asia where less than 20% of the population has access to safe drinking water. Consequently, about 40,000 deaths are reported monthly, mostly infants and children, due to diseases derived from microbiological contamination of water such as diarrhea, cholera, and hepatitis.

In addition, the lack of safe drinking water may arise as a temporary problem in other parts of the world, caused by emergency situations such as natural disasters or armed conflicts. In fact, according to the UNICEF report [2], children under the age of fifteen living in countries affected by protracted conflict are on average almost three times more likely to die from diarrhoeal diseases caused by a lack of safe water than by direct violence.

Because this is a problem associated with low-income countries or emergency situations, a low-cost, simple, and sustainable solution is needed. In this sense, Solar water disinfection (SODIS) is an inexpensive Household Water Treatment (HWT) approved by the WHO that does not require the use of chemical disinfectants, exemplifying its sustainability. This solution is based on the ability of UV radiation from sunlight to inactivate pathogens that are present in water, such as bacteria, viruses, and protozoa, the effectiveness of which has been undoubtedly demonstrated over several decades [3]. Moreover, it presents an optimal solution for low-income countries, as these regions usually have high solar irradiance levels throughout the year.

In the SODIS method, microbiologically contaminated water is introduced in transparent plastic containers, such as polyethylene terephthalate (PET) bottles [4] or polyethylene (PE) bags [5], as we explain in detail in Section 2, then exposed to direct sunlight until enough UV radiation is received to inactivate the pathogens. This amount of radiation is known as the “lethal UV dose” (expressed as W · h/m${}^{2}$), and depends on the properties of the water, the level and type of the microbiological contamination, and the characteristics of the SODIS containers (transmittance to UV light, size, and shape) [6]. There are several studies in the literature [7,8,9,10] that have established methods for estimating the lethal UV dose for different type of SODIS containers and pathogens.

However, one of the main drawbacks of the SODIS method is that there are no practical solutions to determine when the lethal UV dose has been reached due to the great restrictions imposed by the SODIS method. On the one hand, because the purpose of the solution is to complement the low cost of SODIS containers, its price cannot increase the total cost very much. On the other hand, because a high commitment to the environment is needed in this type of project, an eco-sustainable solution should be employed. As we show in Section 2, several approaches have been proposed in the literature to determine the end of the disinfection process; however, these do not strictly accomplish both constraints, and as such cannot practically be applied in real situations.

In fact, the currently used solution is not to measure the UV radiation received; rather, it is to wait a reasonable time to inactivate the pathogens, that is, to expose the SODIS containers for at least 6 h under full sunshine or 48 h on cloudy conditions [11]. However, this solution makes inefficient use of the valuable resources provided by SODIS, as an overly extended exposure time is used to ensure that the water has been disinfected, when in fact the lethal UV dose may have already been reached well before.

In this work, we propose a low cost and eco-sustainable electronic device to be used in conjunction with SODIS containers to determine the end of the disinfection process on the basis of the corresponding lethal UV dose. The proposed solution strictly accomplishes the required restrictions imposed by the SODIS method. We design a low cost device (around EUR 12) which has enough autonomy to work for months with small low cost disposable batteries, thereby avoiding the use of materials that are considered hazardous waste at the end of their useful life, such as rechargeable batteries. In addition, the physical design of this device is valid for any region and type of SODIS container, as the lethal UV dose can be programmatically changed in each case according to the relevant literature.

The rest of this paper is organized as follows. Section 2 describes related works in the literature, while in Section 3 a general overview of the methodology that has been applied for the design and implementation of the proposed device is presented. After that, the different steps of this methodology are described in detail in the following sections; specifically, an analysis of different low cost UV sensors is summarized in Section 4 in order to select the most accurate one, while Section 5 describes the design of the final device and its testing in real conditions. Finally, Section 6 summarizes the main conclusions of this paper.

## 2. Related Work

As mentioned above, in the SODIS method transparent plastic containers are filled with microbiologically contaminated water and exposed to direct sunlight to disinfect it. This requires that two questions be answered, namely, the material used for the containers and how to determine the end of the disinfection process.

With respect to the first question, we can find in the literature a great variety of works that propose and analyze the effectiveness of different type of materials. Traditionally, transparent polyethylene terephthalate (PET) bottles have been widely used due to their efficient transmission of UV-A radiation (about 85–90 percent) [4]. However, this type of material does not transmit UV-B radiation [11], which produces the most powerful genome damage to viruses and bacterial pathogens through direct photo-inactivation mechanisms [7,12]. In addition, there are concerns about the migration of chemical contaminants from PET bottles into water, although there is no direct scientific evidence of this [13]. Thus, more efficient and safer materials have been studied and successfully evaluated in the literature, such as polystyrene (PS) [14], polycarbonate (PC) [7], polymethylmethacrylate (PMMA) [15], and polyethylene (PE) [5]. In particular, the use of PE bags [6] has been viewed as a promising solution, as this material has good transmission of UV-B radiation. Moreover, the bags can be easily delivered and distributed when empty because they are softer and more flexible than PET bottles.

Regarding the second question, as mentioned in Section 1, the end of the disinfection process is reached when enough UV radiation is received to inactivate the pathogens that are present in the water, which is defined as the lethal UV dose. In this sense, several solutions have been proposed in the literature to measure the accumulated UV irradiation. In [16], an electronic control system based on an UV-A photodiode was included as part of a SODIS batch photo-reactor, consisting of a glass tube positioned at the focus of a compound parabolic collector (CPC) mirror and two reservoir tanks (one for the untreated water and other for the treated water). In this framework, the control system is used to control the water flow between the tanks by means of electronic valves. However, the proposed dosimeter has two main drawbacks: on the one hand, it is integrated into the batch photo-reactor, which is a more expensive solution than the use of classic plastic bottles and bags; thus, it is not designed to accomplish the required constraint of being a low cost device. On the other hand, it can only measure the amount of UV-A radiation, not the UV-B radiation. Similar approaches can be found in [17,18], where, although in these cases UV-B radiation has been taken into account, the proposed solution is integrated in a CPC reactor system.

In [5], a chemical solution was proposed based on compounds that change color after receiving the appropriate UV dose. Although it is a very inexpensive approach, the main problem of this solution is its sustainability. Because these dosimeters are disposable one-use systems, they need to be replaced with each SODIS application, which implies supply chain issues.

Finally, there are electronic devices which work in a similar way to our proposed approach. One of the most popular is marketed under the name “WADI” [19]. It consists of a UV meter powered by a solar panel, which can be placed alongside the SODIS containers to determine when solar water disinfection has been reached. The main drawback of this device is its price (around USD 30), which does not accomplish the required low cost constraint, preventing its use in low-income countries. The presumed advantage of this device compared to ours is its lifetime; it can be theoretically used for several years, as it does not need to change the battery. However, this is not true in practice, because a recalibration of the UV sensors is needed every 12–24 months in order to maintain their accuracy [20,21]. Thus, the useful lifetime of a similar device should not exceed this period.

Another electronic device, known as “SCIPIO” (Scientific Purification Indicator), was proposed in [22] to measure the accumulated UV irradiation in SODIS containers. In this case, the device is designed to be introduced into the SODIS container during the disinfection process. As in the previous approach, the main drawback of this solution is its price. Although the authors do not provide the manufacturing cost, there is no doubt that this device cannot accomplish the required low cost constraint, as it integrates a UV sensor, daylight sensor, temperature sensor, gyroscope, capacitance change sensor, memory LCD, Bluetooth, and solar power supply.

## 3. Methodology

In this paper, we propose an electronic device to determine the end of the disinfection process in SODIS bottles and bags that is able to solve the problems with the previous approaches discussed above. The operation of the device is very simple: when SODIS containers are filled with contaminated water and exposed to sunlight, the device should be placed next to these containers in order to measure the accumulated UV irradiation, providing notice when the lethal UV dose has been reached. Because this dose may be different in different regions and types of SODIS containers, the proposed device must be programmable. In this way, the same electronic device can be mass-produced, reducing the manufacturing cost, and the corresponding lethal UV dose can be set through software depending on the region and the characteristics of the employed SODIS containers.

According to these requirements, we propose a microcontroller-based device which includes a UV radiation sensor to capture the solar irradiance corresponding to the UV-A and UV-B spectrum ranges (365 ± 10 nm and 330 ± 10 nm, respectively). The microcontroller is programmed to calculate the accumulated UV irradiance, initialized by a pushbutton, which is then compared with the lethal UV dose. When this value is reached, the device provides notice that the disinfection process has been concluded by means of an LED. A power supply stage with sufficient autonomy to work for months is included, and uses batteries that are not considered hazardous waste at the end of their useful life in accordance with the commitment to environmental responsibility required for this project. In addition, a low battery indicator light is included in order to prevent the use of the device in such a state, as notification of the end of disinfection may no longer be reliable.

The most challenging constraint of this project is the need for a low cost solution, which imposes severe restrictions on the design of the device and selection of the electronic components. In addition, low energy consumption is another important limitation that must be taken into account. The methodology applied for the design and implementation of the proposed device is summarized as follows. First, we analyzed the performance of several representative low cost UV radiation sensors in order to select the most accurate one. To do this, we developed a test prototype based on the Arduino platform to compare the response of these sensors with a reference pattern, as described in Section 4. After selecting the most suitable UV sensor, the next step was to design the final microcontroller-based device while accomplishing the low cost and sustainability constraints, as shown in Section 5. Finally, the device was manufactured and tested in real conditions to analyze its accuracy by comparing it with a reference pattern.

## 4. Analysis of Low Cost UV Sensors

### 4.1. Tested Sensors

For this analysis, we selected three low cost sensors able to measure both UV-A and UV-B radiation.

*GUVA-S12SD UV-A* [23]: this sensor, which is based in Schottky technology, can work in the spectral range from 240 nm to 370 nm, with maximum responsiveness at 350 nm. In particular, we chose the “Analog UV Light Sensor Breakout—GUVA-S12SD” board from Adafruit [24], which integrates this sensor, as well as a preamplifier stage to impose the operating voltage level. This board is shown on the left in Figure 2.*LAPIS ML8511* [25]: this sensor works in the spectral range from 280 nm to 390 nm, with maximum responsiveness at 365 nm. In this case, the sensor integrates its own preamplifier stage, which imposes an operation voltage from 2.7 V onwards. The “SEN0175” board from DFRobot [26], shown in the center of Figure 2, was chosen.*Vishay’s VEML6075* [27]: this sensor works in the spectral range from 315 nm to 375 nm, and can separately detect UV-A and UV-B radiation, with maximum responsiveness at 330 nm and 365 nm, respectively. It is based on CMOS technology, integrating a photodiode, amplifier, and analog/digital converter on a single chip. Reading is performed using the I2C protocol, and it operates within a power supply range of 1.7 V to 3.6 V. In this case, we chose the “PIM460” board from Pimoroni [27], shown on the right in Figure 2.

### 4.2. Sensor Data Collection

In order to obtain the UV irradiation values from the selected sensors, we developed a testing device based on the Arduino UNO platform [28], which includes a data-logger module with a mini-SD card to save these data as files [29]. In addition, an external Real-Time Clock module [30] is employed to register the precise time at which each value is captured, as these data must be compared with a reference pattern and both should be synchronized. Figure 3 shows the schematic of the developed testing device.

This testing device was programmed to register data samples captured by the sensors every 10 s, which is the sampling frequency used in the pattern device. Within this interval, three different measures are saved for each sensor: the minimum value, the maximum value, and the mean value.

Before saving them, the values provided by the sensors must be transformed into irradiance values (in W/m^2^) using the manufacturer’s data sheets and recommendations. It should be noted that these values are considered raw data until final calibration of the sensors is carried out. In the case of the Adafruit GUVA-S12SD, the voltage $V_{o}$ provided by the board is read through an Arduino analog pin and converted into an irradiance value using the following relationship [23,24]:(1)$$Irradiance\left(W/m^{2}\right)≅0.1006\cdot V_{o}$$

In a similar way, the irradiance value can be calculated from the voltage $V_{o}$ provided by the DFRobot ML8511 board as follows [25,26]:(2)$$Irradiance\left(W/m^{2}\right)=0.384\cdot V_{o}-77.749$$

Finally, the Pimoroni VEML6075 module provides a digital output through I2S communication. In this case, digital values related to UV-A and UV-B irradiance are obtained using the Arduino library provided by the manufacturer. These digital values can be employed to obtain the UV irradiance as follows [31]:(3)$$Irradiance\left(W/m^{2}\right)=uva\cdot 5.376\cdot 10^{-3}+uvb\cdot 2.381\cdot 10^{-3}$$

### 4.3. Sensor Selection and Calibration

In order to evaluate the tested sensors, we compared the data collected from them with the corresponding reference values captured by a radiometer (our reference pattern). A CUV 5 radiometer from Kipp and Zonen Company was employed [32]; it is able to measure radiation between 280 nm and 400 nm, corresponding to the UV-A and UV-B spectrum. This radiometer has its own portable data logger called METEON [33], configured to store data every 10 s; during this frame, a sample with the maximum, minimum, and integral values was saved.

To build our dataset, we collected irradiation data from the tested sensors and the radiometer for three different days (9–11 March 2021 ). Data were collected over the whole daylight range, from sunrise to sunset in Spain (approximately 8 a.m. to 7:30 p.m.) in order to include both very low irradiation data (at sunrise and sunset) and very high irradiation data (during the middle hours of the day). In addition, the weather on these days was changeable, alternating between cloudy and clear skies. Because data were stored every 10 s, a total of 12,420 samples were collected, including a wide range of irradiation levels.

The data collected on the first day were used to select the most accurate sensor by analyzing the correlation between the values provided by the sensors and the values provided by the radiometer (maximum, average, and minimum values). The data collected on the second day were used to obtain the calibration parameters for the best sensor, and the data collected on the third day were used to evaluate its calibration.

Figure 4 shows a graphical representation of a selection of the data captured on the first day. It can be seen that the response provided by all the sensors is very similar to the pattern provided by the radiometer.

In order to quantify the degree of linear dependence between each sensor and the pattern, a correlation analysis was performed. As shown in Table 1, all the tested sensors had a high degree of correlation, with the highest value corresponding to the GUVA-S12SD sensor (Adafruit module), which achieved close to 96% correlation. Thus, this sensor was selected for integration into the final device.

After the sensor had been selected, the next step was to calibrate it. As mentioned above, the data collected on the second day were used to obtain the calibration parameters (i.e., the regression line). First, we removed the outliers in the dataset, analyzing the atypical values according to the distribution of the data. In Figure 5, the relationship between the sensor measures and the pattern after filtering out atypical values can be observed. From these data we obtained the corresponding regression line that should be applied to calibrate the sensor; the equation is shown in Figure 5.

In order to evaluate this calibration, we then applied the obtained regression line to the data collected on the third day, which were not used in the calibration process. In Figure 6, the calibrated values of the sensor are compared with the reference pattern provided by the radiometer, showing the goodness of the proposed calibration.

In order to carry out a statistical evaluation of the calibration, we calculated the absolute error of the samples with respect to the reference pattern, ordered the samples according to this absolute error, and looked for the first sample in which the error exceeded 5%. This occurred at the 90.29th percentile, i.e., only 9.71% of the calibrated values exceeded an absolute error of 5%, which demonstrates the excellent response of the selected sensor.

## 5. Device Design and Validation

With the most suitable UV sensor for our project selected, in this section we describe the electronic design of the device, with special emphasis on the choice of components and their physical implementation on a PCB (Section 5.1). Likewise, both power consumption (Section 5.2) and cost (Section 5.3) are analysed in order to verify that the restrictions imposed in the project are met. Finally, we describe the tests carried out to evaluate the real operation of the device (Section 5.4).

### 5.1. Electronic Design and Implementation

As previously mentioned, our proposed programmable microcontroller-based device must be capable of calculating the accumulated UV radiation and providing a notification when the lethal dose has been reached. For this purpose, the device has been divided into the following functional stages: control (Section 5.1.1), power supply and regulation (Section 5.1.2), sensor adaptation (Section 5.1.3), and programming and RESET (Section 5.1.4). All these functional stages are integrated on a PCB board, as described in Section 5.1.5.

#### 5.1.1. Control Stage

This stage consists of the microcontroller and two very low power LEDs (along with their respective bias resistors), which are used indicate the end of the disinfection process and low battery status, as shown in Figure 7. A button is added to the PB2 pin to make the operation mode of the device easily understandable and intuitive. Thus, when the user presses the button, if the operation LED flashes for 3 s, the disinfection process is not finished yet; if the operation LED stays on for 3 s, the disinfection process has finished. In addition, this same button is added to pin PB5 through an RC circuit in order to make a hard reset easy by pressing it for a period of 3 s. Regarding the other LED indicator, it remains off in normal operating mode, and only lights up when a critical battery level is reached, indicating that it is advisable to stop using the device.

In order to improve the power consumption of the device, when the disinfection process is over the device goes into “sleep” mode (ultra-low consumption mode) until the user realizes it and deactivates it, turning the switch to off mode. Furthermore, the device enters this state when there is an absence of UV radiation.

For selection of the microcontroller, because sampling frequency was not a determining factor we analyzed only those devices with low cost, small size, and capacity to work at low voltages (See Table 2).

From these, the ATtiny85V was chosen on account of its balanced consumption and the possibility of configurating it in different sleep modes, being able to consume a minimum of 0.1 μA in this state. It is composed of six programmable I/O pins, with two PWM channels, a 10-bit A/D converter, and several communication protocols. The complete schematic can be reviewed in the manufacturer’s datasheet [34].

As can be seen in Figure 7, the power supply of this microcontroller was set at 1.8 V, which is supplied by the regulation stage described in the next section.

#### 5.1.2. Power Supply and Regulation Stage

In a project such as this, where the device must be exposed to the sun for a long time, it might be thought that a good power supply system would be to use, for example, a photovoltaic panel connected to LiPo rechargeable batteries or supercapacitors. However, this type of solution was discarded in our case because it cannot accomplish the requirements needed by the device. On the one hand, due to recent changes in legislation, batteries containing lithium are now considered as hazardous waste at the end of their life-cycle [35]. Therefore, the use of rechargeable batteries was not considered in this project. On the other hand, although the use of supercapacitors would satisfy the eco-sustainability requirement, their price (around EUR 5–6 [36,37,38], together with the photovoltaic panel at around EUR 3 [39,40,41]) would excessively increase the cost of the device. As we show in Section 5.3, the total cost of the proposed device is around EUR 12, which would be increased by EUR 7–8 if a solution incorporating a photovoltaic panel and supercapacitors were to be used, representing an increase of around 60–70%.

Instead, we used single-use Zinc alkaline batteries, which accomplishes the eco-sustainability conditions in addition to having a reduced cost and high charging capacity, as can be seen in Table 3. Because the chosen microcontroller can operate with voltages starting from 1.8 V, we used two batteries in series, achieving a nominal voltage of 3 V.

Figure 8 shows the diagram of the power supply and regulation stage. The V_BATT signal consists of 3 V supplied by two AA batteries placed in series using a battery holder, which is connected to switch S1 to turn the device on and off. After this, the signal is split into two: the battery level measurement signal (“BATT_LEVEL” signal), which is discussed later, and the voltage regulation stage signal. The latter consists of a protective Schottky diode followed by a MIC5225-1.8YM5 voltage regulator, which provides a stable power supply to the rest of the circuit. Specifically, this regulator provides a stable voltage of 1.8 V for input voltages of at least 2.3 V, which, as mentioned before, is the minimum with which the microprocessor can operate. The connections in the regulator were implemented following the recommendations provided in the manufacturer’s datasheets [42].

On the other hand, the “BATT_LEVEL” signal is the input to the battery level measurement stage, shown in Figure 9. This stage consists of a simple asymmetrical buffer with a rail-to-rail operational amplifier, for which the input is a voltage divider that allows for adapting the battery level (up to 3 V) to a level suitable for analog reading by the microprocessor, which is powered at 1.8 V. The output signal “PB/LOW_BATT” is connected to one of the inputs of the microprocessor, as shown in Figure 9, and determines when the low battery indicator LED should blink.

#### 5.1.3. Sensor Coupling Stage

The conditioning circuit of the signal from the GUVA sensor was designed according to the recommendations provided by Adafruit under the open-source license [24], in this case using an asymmetrical power supply of 1.8 V for the operational amplifier, as shown in the diagram in Figure 10.

#### 5.1.4. Programming and RESET Stage

An external push button was included in the design, which has three functionalities. First, it allows the microcontroller to be set in read mode in order to be programmed by entering the necessary code via ISP. Second, by holding the button down for 3 s the user can reset the reading of the accumulated UV dose in case multiple disinfection measures with the SODIS containers need to be made during the day. Finally, with a simple touch the user is able to check the state of the disinfection process.

Its design is the typical switching circuit shown in Figure 11, consisting of an external protection diode, pull-up resistor, and a filter capacitor to avoid the rebound effect. Under normal conditions, it offers a HIGH voltage level at the output (PB2/RESET) and a LOW level when the push button is pressed.

#### 5.1.5. PCB Board Design

For the manufacture of the final device, a double-sided PCB design was chosen in order to adapt to a commercial housing (CU-1941) [43] in a tight fit. For protection, most of the electronic components are placed inside the housing (bottom side), leaving only the ON/OFF switch, the UV sensor, the LED indicators, and the push button on the outside (top side). After the device has been assembled, an epoxy resin coating is added to the top side of the device in order to protect it from the impact of operational conditions in real environments (i.e., sun exposure, presence of dust). The final PCB is shown in Figure 12.

With regard to the design parameters using routing software, a GRID was chosen according to the PCB manufacturer’s capacity and width and the thickness of tracks and vias according to the IPC-2221A standard.

### 5.2. Power Consumption

From the very beginning, consumption was an influential factor in the design and development of the device, as the dosimeter needed to be able operate for several months with two AA batteries as the only source of energy. The estimated power consumption of each of the important components of the device is, according to their data sheets, as follows:OPAMP MCP6001: 100 μA [44]OPAMP AD8515: 300 μA [45]ATtiny85V microcontroller: 280 μA (at 1.8 V and 1 MHz) [34]APTD2012LSURCK LED diode: 2 mA [46]MIC5225-1.8 V regulator: 70 μA [42]

With these data, and taking into account that the disinfection LED indicator only works when the user presses the button, there are a number of energy consumption modes:-Operating normal mode (without LEDs): 750 μA-Verification mode with disinfection process not finished (LED blinking by PWM 45% of the time for a period of 3 s): 1.65 mA-Verification mode with disinfection process finished (LED fully operational for a period of 3 s): 2.75 mA-Sleep mode (ultra-low consumption): 0.1 μA

In order to verify these theoretical values, we empirically measured the power consumption of the device, obtaining the following data:LED on: 1820 μALED off: 910 μALED flushing: 1320 μA

Using this empirical power consumption, we can calculate the average consumption considering a worst-case scenario in which the user pushes the button ten times per hour and the disinfection process finishes in 3 h: (4)$$\begin{array}{cc} \mathrm{Avg}.\;\mathrm{consumption} & =\frac{29\cdot 3\;s\cdot 1.32\;mA+1\cdot 3\;s\cdot 1.82\;mA+(3\cdot 3600\;s-30\cdot 3\;s)\cdot 0.91\;mA}{3\cdot 3600\;s}\\ \; & =\frac{9866.4\;mA\cdot s}{3\cdot 3600\;s}=0.914\;mA \end{array}$$

It can be seen that in bad scenarios the consumption is somewhat similar to the normal operating mode (LED off). Using 3112 mAh alkaline batteries (Table 3) and a consumption value even larger than the calculated one (0.92 mA), the operating time in hours can be estimated as
(5)$$\mathrm{Runtime}\;\left(h\right)=\frac{3112\;mAh}{0.92\;mA}=3382.61\;h$$

Thus, for the hypothetical case in which the UV dosimeter is in operation during all the daylight hours of the day in Spain, which in the extreme case is 3000 h per year, the lifetime of the proposed device is at least 411 days, i.e., 13.5 months of continuous active operation during daylight. In addition, it should be noted that the device enters “sleeping mode” (ultra low power consumption) when the endpoint time of disinfection is been reached or when there is no UV radiation presence. In this way, the device does not need to wait until the user realizes the disinfection process is over. This implies that the lifetime of the device without changing the batteries should be much higher than 411 days in real use.

### 5.3. Device Unitary Cost

Table 4 shows a detailed budgeting of all the components to be used in the case of manufacturing 100 units of the final device. This budget includes PCB fabrication, component assembly, and shipping. To account for fabrication and assembly, a purchasing estimate was carried out on the JLCPCB website [47], using standard options for fabrication along with non-priority shipping.

Taking into account the budget shown, we estimate a unit manufacturing cost of less than EUR 12 for a batch of 100 units.

### 5.4. Device Validation Tests: Error in the Endpoint Time

The objective of these tests is to calculate the difference in the time needed to reach the lethal UV dose between the proposed device and the radiometer used as reference pattern, i.e., the relative error of our device in terms of the time that indicates the end of the disinfection process. To do this, both devices were first exposed to direct sunlight simultaneously over a period of time, then the obtained UV radiation data were recorded. Figure 13 shows the values for both devices. The accumulated UV radiation was then calculated in both cases and the difference in the time required to reach the same values was determined.

Several tests were carried out, for which the device was programmed with different values for the lethal UV dose. In particular, for a lethal UV dose of 45 Wh/m${}^{2}$ (the lethal value corresponding to enterococcus), which is the most widely used in practice, the proposed device reached this value with a delay of 40 s with respect to the radiometer after approximately 3 h of measurement. According to these values, our device obtains a relative error of 0.36% for the time that indicates the end of the disinfection process. In addition, this endpoint time is delayed with respect to the real time, meaning that it does not imply any risk that might affect the disinfection of the water. On the contrary, the later indication introduces a small safety margin that eliminates the risk of drinking water being unsuitable for consumption. These results are shown in Figure 14.

## 6. Conclusions

In this paper we have proposed a low cost and eco-friendly sustainable electronic device that can be used to determine the end of the disinfection process in the SODIS method. The device should be placed next to SODIS containers in order to measure the accumulated UV irradiation, notifying users when the lethal UV dose has been reached. Because this lethal UV dose may be different for different regions of the world and different types of SODIS containers, a programmable device has been designed that can be used in any case. Because the DNA repair mechanisms of exposed cells are overwhelmed by UV radiation [11,48], the damage induced by sunlight continues even when the cells are taken out of the sun and incubated in the dark. This is particularly important with respect to the possibility of bacterial re-growth during storage.

The most challenging constraint of this project was to develop a low cost solution, which imposed severe restrictions on the design of the device and the selection of the electronic components. First, as mentioned in Section 4, we have analyzed different low cost UV sensors in order to select the most accurate one by comparing their response with the reference pattern provided by a radiometer. This analysis shows that low-cost sensors provide very good responsiveness in the measurement of both UV-A and UV-B radiation, reaching a correlation with respect to the reference pattern of around 96% in the case of the GUVA-S12SD sensor.

In Section 5, we have described the electronic design of the proposed device, which uses the selected sensor to measure the accumulated UV radiation and compares this value with the lethal UV dose to determine the end of the disinfection process. In addition to the low cost constraint, sustainability has been taken into account in this design as well, leading to a proposed device which has sufficient autonomy to work for more than a year with small and low-cost disposable batteries, thereby avoiding the use of rechargeable batteries, which are considered hazardous waste at the end of their useful life. Finally, the device was manufactured and tested in real conditions by comparing its response with the reference pattern provided by the radiometer. According to these tests, a relative error of only 0.36% in the endpoint time was obtained.

## Figures and Tables

**Figure 1 sensors-23-00575-f001:**
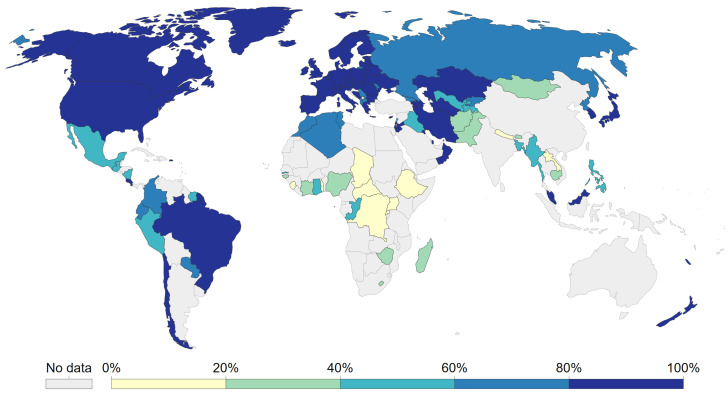
Share of the population with access to safely managed drinking water in 2020 [1].

**Figure 2 sensors-23-00575-f002:**
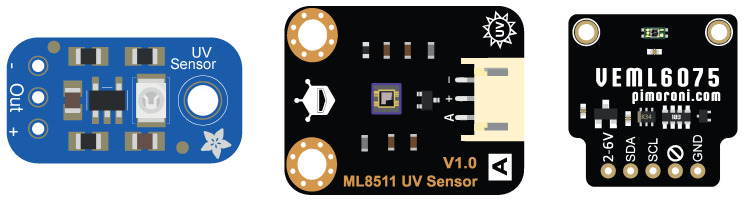
Boards integrating the chosen sensors: GUVA-S12SD UV-A (**left**), LAPIS ML8511 (**center**), and Vishay’s VEML6075 (**right**).

**Figure 3 sensors-23-00575-f003:**
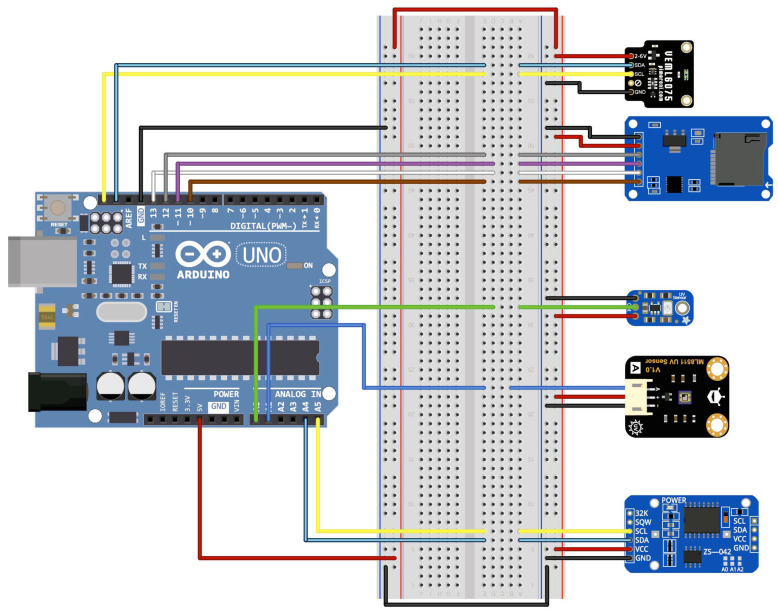
Schematic of the developed testing device.

**Figure 4 sensors-23-00575-f004:**
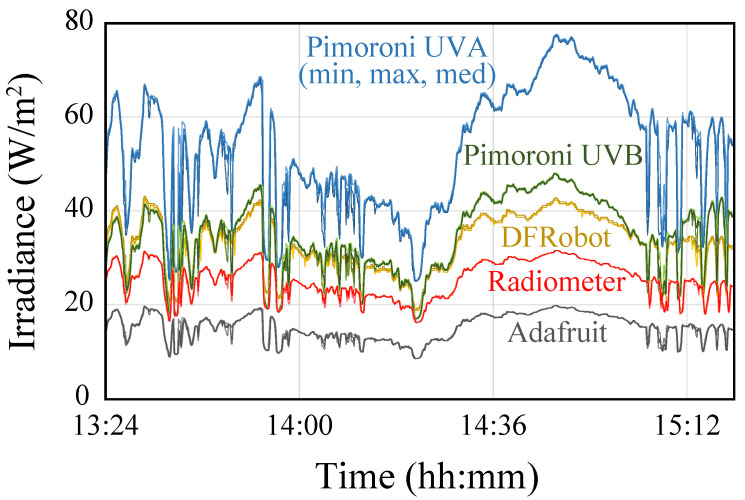
Graphical representation of the data captured by the three sensors and the radiometer on the first day.

**Figure 5 sensors-23-00575-f005:**
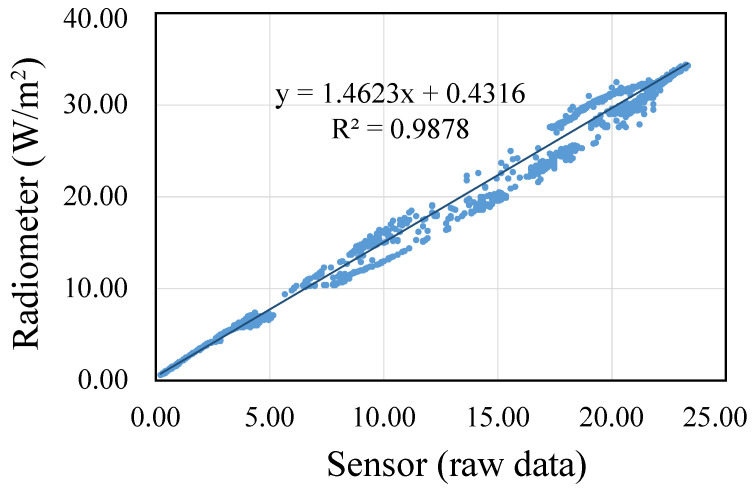
Relationship between the sensor measures and the pattern provided by the radiometer.

**Figure 6 sensors-23-00575-f006:**
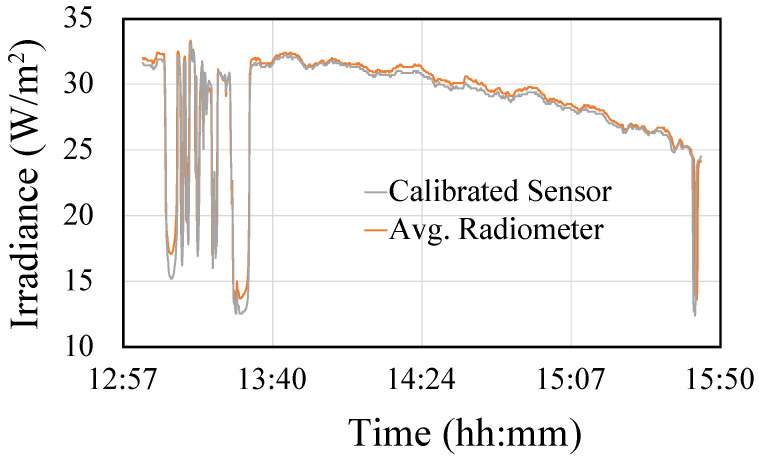
Comparative between the calibrated values of the sensor and the reference pattern provided by the radiometer.

**Figure 7 sensors-23-00575-f007:**
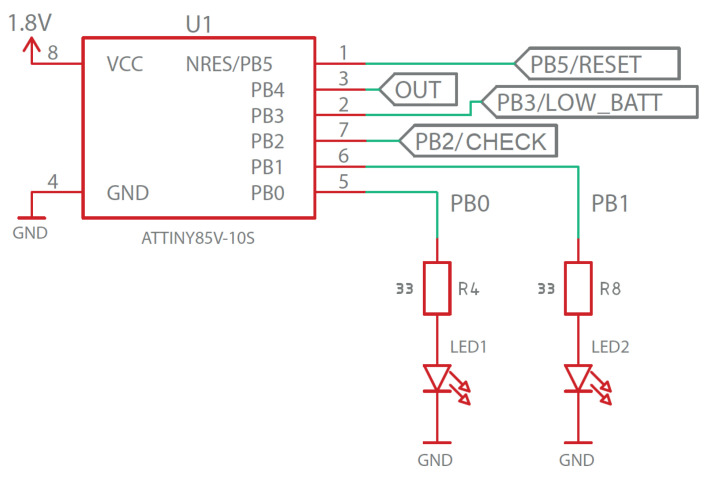
Control stage schematic design.

**Figure 8 sensors-23-00575-f008:**
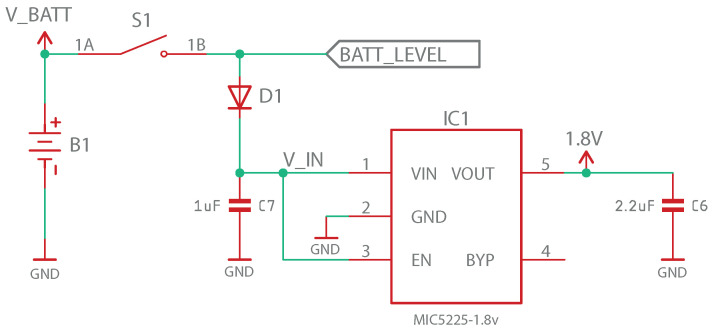
Power supply stage schematic design.

**Figure 9 sensors-23-00575-f009:**
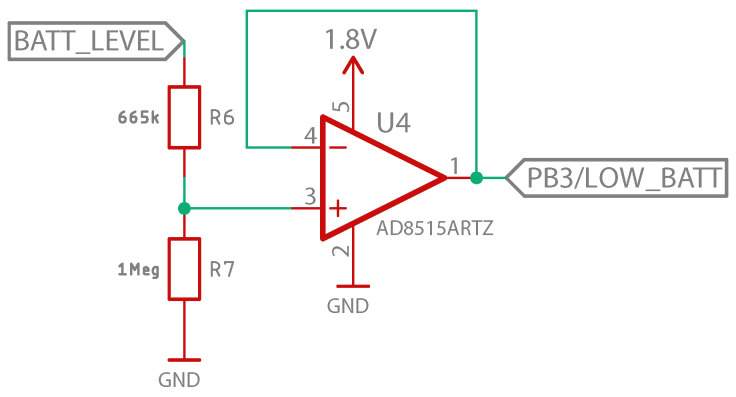
Battery level meter stage schematic design.

**Figure 10 sensors-23-00575-f010:**
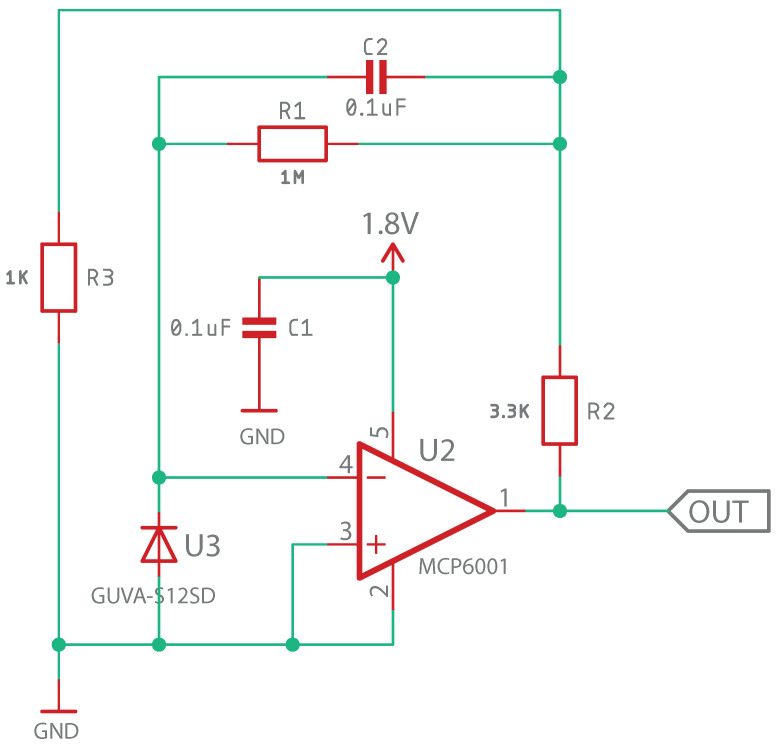
UV Sensor coupling stage schematic design.

**Figure 11 sensors-23-00575-f011:**
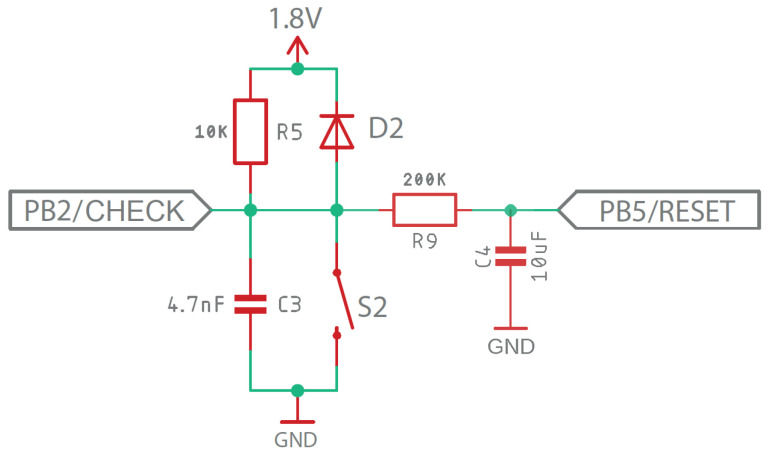
Switch button schematic design.

**Figure 12 sensors-23-00575-f012:**
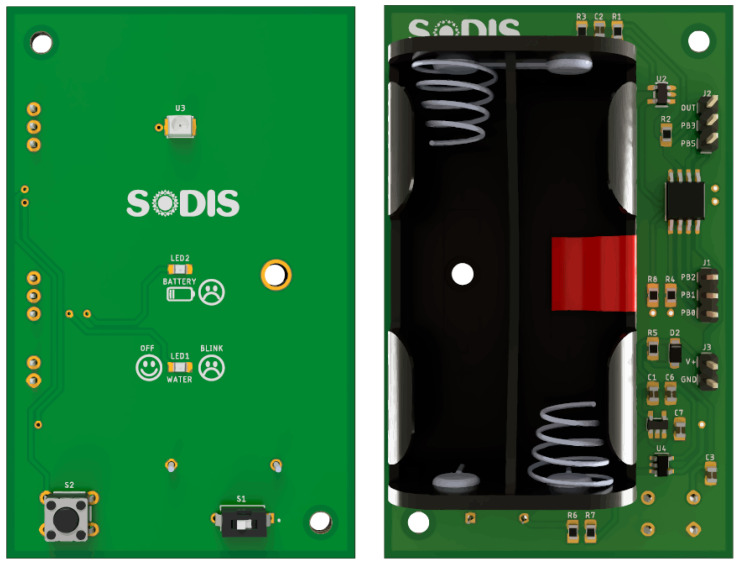
PCB device: top (**left**) and bottom (**right**).

**Figure 13 sensors-23-00575-f013:**
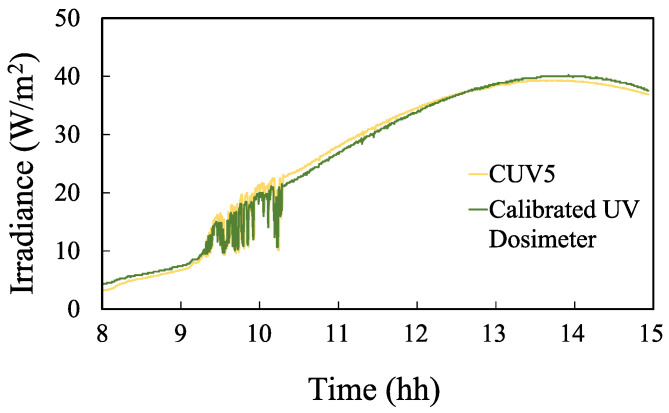
Irradiance (W/m${}^{2}$) measured on 30 August 2021: proposed device (green) vs. CUV5 radiometer (yellow).

**Figure 14 sensors-23-00575-f014:**
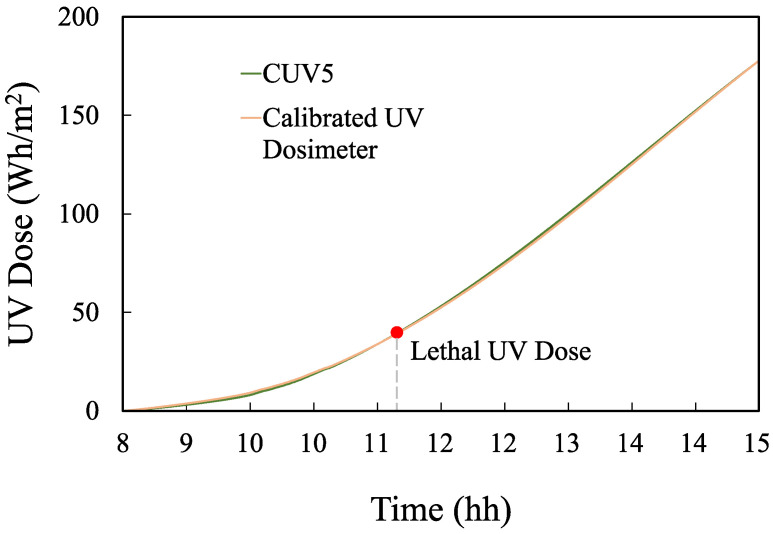
UV dose (Wh/m${}^{2}$) measured on 30 August 2021: proposed device (orange) vs. CUV5 radiometer (green).

**Table 1 sensors-23-00575-t001:** Correlation degree of the data captured by the sensors and the pattern reference provided by the radiometer.

	Radiometer		
	**Min.**	**Max.**	**Avg.**
Adafruit_min	0.9461	0.9244	0.9427
Adafruit_max	0.9301	**0.9600**	0.9503
Adafruit_med	0.9506	0.9526	**0.9584**
DFRobot_min	0.9410	0.9185	0.9372
DFRobot_max	0.9271	0.9577	0.9477
DFRobot_med	0.9475	0.9496	0.9555
Pimoroni_UVA_min	0.9094	0.8828	0.9036
Pimoroni_UVA_max	0.8934	0.9138	0.9094
Pimoroni_UVA_med	0.9126	0.9077	09174
Pimoroni_UVB_min	0.8871	0.8558	0.8796
Pimoroni_UVB_max	0.8693	0.8868	0.8846
Pimoroni_UVB_med	0.8899	0.8809	0.8933

**Table 2 sensors-23-00575-t002:** Initial list of microcontrollers.

Model	Price	Consumption	Pins	Voltage	Frequency
ATmega328P	1.50 €	240 μA	28	1.8–5.5 V	0–16 MHz
ATtiny85	1.05 €	300 μA	8	2.7–5.5 V	0–20 MHz
ATtiny13V	1.10 €	240 μA	8	1.8–5.5 V	0–10 MHz
ATtiny85V	1.00 €	300 μA	8	1.8–5.5 V	0–10 MHz

**Table 3 sensors-23-00575-t003:** Initial list of alkaline batteries.

Model	Form Factor	Rated Voltage	Life (mAh)	Material	Price/un
PX1500	AA	1.5 V	3.112	Zn-$MnO_{2}$	0.85 €
4006	AA	1.5 V	2.950	Zn-$MnO_{2}$	0.85 €
PC1500	AA	1.5 V	3.016	Zn-$MnO_{2}$	0.90 €
MN1500	AA	1.5 V	3.016	Zn-$MnO_{2}$	1.50 €

**Table 4 sensors-23-00575-t004:** Budget and total cost of manufacturing 100 UV dosimeter units.

Index	Qty	Mfrs Name	Unit Cost	Total Cost
1	100	2462RB	1.17 €	117.01 €
2	100	AD8515ARTZ-REEL7	0.55 €	54.57 €
3	200	APTD2012LSURCK	0.17 €	34.94 €
4	100	ATTINY85V-10SU	1.03 €	103.36 €
5	100	CU-1941	1.40 €	139.90 €
6	100	GUVA-S12SD	3.40 €	345.10 €
7	100	MCP6001T-I/OT	0.19 €	19.08 €
8	100	MIC5225-1.8YM5-TR	0.31 €	30.52 €
9	200	LR6 G07(2S)	0.26 €	51.82 €
10	-	Electronic fungibles		164.62 €
			Subtotal	1060.92 €
		Manufacturing		25.12 €
			Subtotal	1086.04 €
		Assembly		93.44 €
			Subtotal	1176.48 €
		Shipping		17.40 €
			Total	1196.88 €

## Data Availability

The data presented in this study are available on request from the corresponding author.

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
