# Peer review of "A Low Cost and Eco-Sustainable Device to Determine the End of the Disinfection Process in SODIS"

_sensors, 2023, doi:10.3390/s23020575_

Round 1

Reviewer 1 Report

Solar water disinfection (SODIS) is widely used way to inactivate the pathogens. In this paper, the authors developed a reliable and a low-cost sensor to determine the end of the disinfection process. This study is very interesting and has high applicational value. The work is well presented. And the related previously reports have been well reviewed and a systematical comparison study has been carried out. I would like to recommend it for publication in Sensors in its current form.

Author Response

We appreciate your generous words. There have been many hours of hard work. Best regards.

Reviewer 2 Report

This manuscript detail the development of an electronic device to determine when the lethal UV dose has been reached in Solar water disinfection (SODIS) containers, which is low-cost and has enough autonomy to work for months with small low-cost disposable batteries.  In their approach, the authors first analyzed different low-cost UV sensors in order to select the most accurate one by comparing their response with a reference pattern given by a radiometer. The device has been manufactured and tested in real conditions to analyze its accuracy, obtaining satisfactory results. 

Dear authors, Excellent work. The manuscript is outstanding and well-organized. The experimental results to meet the research objectives are complete, well-presented, and thoroughly discussed. After making a few minor changes, the manuscript is ready for publication.

1. The manuscript did not evaluate the measurement precision of the new device. Is it defined?

2. In this regard, the conclusion presents "...accuracy at the last endpoint...". Please clarify, by which statistical criteria was the accuracy of measurements assessed?

Author Response

We really thank you for your generous words and for the time you have spent in the review process. Next lines, we have tried to answer the questions:

1.- The manuscript did not evaluate the measurement precision of the new device. Is it defined?

The precision of the measurements has been defined in terms of the time differences between the disinfection end point reached by the pattern and the device. In this sense, we have included a test on a day where the weather conditions were changeables, in order to validate its reliability in a bad scenario (as can be observed in figure 13).

2.- In this regard, the conclusion presents "...accuracy at the last endpoint...". Please clarify, by which statistical criteria was the accuracy of measurements assessed?

We are sorry to comment that we can not find this sentence,  "...accuracy at the last endpoint...", in the manuscript. So, we don’t know what the reviewer is referring to. Maybe, we have answered this question above. However, we would be pleased to clarify any additional questions.

Reviewer 3 Report

Dear Authors,

Comment 1. The paper emphasizes low cost of the solution for UV dose detection, however, another important aspect is reliability of the circuit that indicates if the water is drinkable or not. Adding some discussion about reliability could be valuable, including needed field trials for qualification the device for practical mass use.

Comment 2. Once the disinfection process is over and the water is drinkable, for how much time it remains drinkable if not exposed to UV ? How the counting of this time is taken into account in the device design ?

Comment 3. According to rows 283-288 (operation description), no lighting/blinking LEDs is supposed to mean that UV sterilization is completed. How it is distinguished from the state that device doesn’t work at all ?

Comment 4. Ecological considerations of avoiding usage of rechargeable Li batteries (rows 301-307) are clear. However, the rechargeable power source (paired to solar panels) could be of significantly smaller capacity and size than batteries. For example, supercapacitor could be considered. The discussion about different rechargeable sources could benefit the paper.

Comment 5. It looks that LEDs are very significant power consumers. However, most of the operation time (blinking), the user would hardly need to see them. Maybe, it is worth to consider adding push button(s) in series to LED(s), so that they blink only when user desires to see them and pushes the button ?

Best regards

Author Response

We really thank you for your generous words and for the time you have spent in the review process. Please see the attachment for the answers at your questions.
